# Exploring the Sustainability Correlation of Value Co-Creation and Customer Loyalty-A Case Study of Fitness Clubs

**Yu-Lung Lee [1] , Lee-Yun Pan [1],\*, Chin-Hsien Hsu [2] and De-Chih Lee [3]**

[1]  Department of Business Administration, National Yunlin University of Science & Technology, Yunlin 64002, Taiwan; yulunglee@hotmail.com

[2]  Department of Leisure Industry Management, National Chin-Yi University of Technology, Taichung 41170, Taiwan; hsu6292000@yahoo.com.tw

[3]  Department of Information Management, Da-Yeh University, Changhua 51591, Taiwan; dclee@mail.dyu.edu.tw

\*  Correspondence: panly@yuntech.edu.tw; Tel.: +886-5-5342601 (ext. 5241)

**Abstract:** Fitness gyms have been gaining popularity among Taiwanese people, which means growing competitiveness for fitness gym operators and the necessity for an examination on how to maintain customer loyalty. Since workouts may require more coaching and interactions than other types of exercise, this study focuses on whether the increased customer interactions and involvement, which may be achieved when the operator works on developing customer value co-creation behavior, can generate increased customer loyalty toward fitness gyms. On the other hand, customers' regular behavior may influence customers' perceived necessity of interactions and involvement; this may in turn influence the correlation of value co-creation behavior and customer loyalty. Accordingly, regular behavior was employed as the moderating variable in this study. The study used SPSS software version 22.0 and AMOS software version 22.0 to evaluate the data collected. By convenience sampling, it distributed questionnaires to 470 subjects, and collected 470 copies, with a return rate of 100%. After eliminating the invalid samples, there were 453 valid samples, with a valid return rate of 96.3%. We distributed questionnaires at outside the fitness clubs in Taichung City from May 20th to June 13th, 2016. The study's results indicate that value co-creation attitude, value co-creation subjective norm, and price affordability have positive effects on value co-creation behavior; value co-creation behavior has a positive effect on customer loyalty, and regular behavior has a negative moderating effect on the influence of value co-creation behavior on customer loyalty.

**Keywords:** Price Affordability; Value Co-creation Behavior; Customer Loyalty; Regular Behavior

## 1. Introduction

Loyalty in business or between businesses and consumers is the key to earning profits [1]. Some studies have pointed out that retaining 5% of old customers can increase corporate profits by 25% to 75% [2]. It costs five times as much to secure a new customer than retaining old ones [3]. As a result, many businesses are eager to build good loyalty within their customer group [4].

IHRSA [5] there are more than 180,000 fitness clubs around the world with an output value of more than $8.4 million. Whether counting by membership or by users, their numbers are growing year by year. The fitness club industry has been valued by all the sectors of the industry, making the competition even more intense. Therefore, the way a business attracts customers is extremely significant to the enterprise. When customers have a high degree of satisfaction from the service or product of the fitness club provide, their loyalty will increase accordingly [6]. In the past, the service

industry offered goods for consumers to make choices, and consumers could only passively accept the arrangement or choose to refuse. There are few opportunities for both parties to interact to get consumers feedback [7], so that they can easily neglect the benefits from the interaction of companies and customers [8]. With fierce competition and diversified choices, customers have more requirements for products quality and services quality. Fitness clubs with high service have items such as compound fitness equipment and professional field equipment for customers. However, in this process, enterprises and customers need to interact and exchange resources to create common value [9]. Customers and businesses also need to maintain mutual aid and dependence to create additional value and reasonable distribution, and improved satisfaction [10].

Different industrial categories have different requirements for customer participation and interaction [11]. In order to meet the needs of both consumers and businesses, the parties need to participate in collaborative interaction to create mutual value [12]. These interactions and participation can improve the service quality [13], perceived value [14], customer commitment [15] and customer satisfaction [16]. In addition to what is stated above, this study's purpose lies in determining whether there are innovative approaches to further improve customer loyalty.

Actively inviting customers to participate in an enterprise value chain and play the role of co-creation can provide customized products and services that are more in line with the customers'needs. At the same time, it will facilitate the development and establishment of a good interaction relationship between enterprises and customers, and will improve the overall operating performance of the company [17,18]. Cossío-Silva [19] pointed out in research that value co-creation has a positive impact on loyalty. Grissemann [20] also found that participation in co-creation has a positive impact on customer satisfaction. Therefore, value co-creation can effectively improve customer loyalty. However, past relevant research has mostly discussed e-commerce [21], innovation, and new product development [22]. Only a few studies have verified the effect of value co-creation in the leisure service industry. Therefore, its capability to increase the customer's loyalty to the fitness club by guiding the customer to engage in value-creating behaviors is the second objective of this study.

It can be determined from past literature that there are many benefits of value co-creation [23], but its ability to make customers engage in the value co-creation of fitness clubs is still in question, as from the perspective of planned behavior, the probability of behavioral occurrence is determined by behavioral intention. The formation factors of behavioral intention include attitude and subjective norms [24]. Attitude refers to the positive and negative evaluation held by an individual for a certain behavior, while subjective norm refers to the approval degree of society, family or friends that an individual feels when he or she undertakes a certain behavior. Therefore, verifying whether the co-creation attitude and subjective norm of value co-creation are the antecedent variable for co-creation behavior. This is the third objective of this study. This is the third objective of this study.

In addition, the price is an important antecedent variable for decision behavior [25]. Customers are influenced by the price's reasonableness or acceptability, which will affect their behaviors in the fitness club [26]. When customers feel that the price offered by the fitness club is reasonable and acceptable, they will be more likely to engage in value co-creation behaviors provided by the fitness club. Therefore, this study believes that price rationality is also an antecedent variable of value co-creation behavior. From the perspective of consumer, consumers' regular exercise behavior may influence their perceived necessity regarding interaction and participation. Compared with consumers who exercise on a regular basis, those who do not may be constrained by a lack of time, thus causing them to exercise fewer than three times per week, less than 30 minutes per time, or without attaining effective exercise intensity during each exercise. These consumers may be less familiar with fitness clubs and may need to seek assistance each time they visit the clubs; thus, they may regard value co-creation as helpful, and become satisfied with value co-creation behavior. Consequently, these consumers are willing to repeatedly purchase the services at the clubs [27], and their loyalty toward the clubs will considerably increase. People who exercise regularly have a relatively more fixed behavioral mode and clearly know what time they should exercise, how long they need to exercise

each time, and which index of exercise intensity they need to achieve. No matter whether this type of consumer is engaged in value co-creation behavior or not, the influences on their loyalty toward fitness clubs are limited. This study incorporates regular behavior as a moderating variable to empirically analyze whether regular behavior generates moderating effects between value co-creation behavior and loyalty. This is the fourth objective of this study.

This study surveys Taiwan's fitness club users, taking the Taiwan fitness club as an example, exploring the study on the relationships between value co-creation behavior, the concept of value co-creation attitude, the subjective norm of value co-creation, price rationality and loyalty. In addition, the study also includes regular behaviors to test whether there will be a moderating effect on co-value behavior and loyalty. This study development of the questionnaire began with a review of the relevant literature.

## 1.1. Loyalty

In the past, scholars extensively have discussed loyalty, providing different definitions of the concept. In a marketing perspective, loyalty combines attitude and behavior, and extends from customer loyalty to employee loyalty [28]. Loyalty is a very important trait that underpins the success or failure of an enterprise [29]. TaghiPourian [30] believes that loyalty is a key factor for the long-term success of enterprises. We consulted the perspectives of Lee [31] and define loyalty as a consumer's behavior in showing an emotional connection and preference toward a fitness club, and a willingness to repeatedly purchase the club services and recommend them to other people.

## 1.2. Value Co-creation Behavior

Consumer needs have changed with the social changes and the rise of the high-tech era in the 21st century, and enterprises need to adjust their internal and external resources and abilities to respond to the changes in the external environments. The relationships between consumers and enterprises have been reorganized, have become interactive, and have demonstrated multiple aspects [32]. Schreier [33] reported that value co-creation means that people from various backgrounds jointly participate in the same event, during which various opinions are gathered together to generate more innovative design ideas. value co-creation is a process where consumers integrate their own knowledge, experiences, and techniques to participate in the products, services, or designs offered by enterprises, thereby modifying and creating new products, while satisfying their own preferences and needs [18]. In their research on service marketing, Vargo [34] revealed that the concept of value co-creation refers to the resource exchange process implemented between consumers and companies, enabling both parties to obtain greater benefits. Value co-creation signifies the high engagement of consumers. Because of those consumers participate, companies have been able to produce highly-customized products and services, which are created based on consumer needs [35]. Payne [10] reported that value co-creation behavior means that collaborative partners face possible challenges and opportunities together, creating value-added behavior through mutual assistance, dependence, and an average and reasonable distribution of resources. We consulted Yi and Gong [36] and define value co-creation behavior as a value mutually determined by enterprises and customers, with both parties pursuing values together and creating added-values through an average and reasonable distribution of resources.

## 1.3. Attitudes toward Value Co-creation

Value co-creation refers to the concept in which an enterprise should regard every customer as the creator of value. Enterprises and customers participate in the value chain and jointly develop new products so that customers can participate in the entire production process to meet the needs of both parties [37]. This is a joint activity between the customer and the enterprise to help one or both parties produce value [38]. Attitude, on the other hand, is an innate quality of learning that enables individuals to have a consistent view of things that they agree or disagree with [39]. It is the evaluation of describable objects and events with the function of guiding individuals to act on

these phenomena and events [40]. Kotler [41] views attitude as an individual's persistent cognitive assessment of emotional feelings for, direction of, and actions towards liking or disliking certain subject matter or concepts. Therefore, an individual will develop positive attitudes toward value co-creation when he/she considers that such value co-creation can bring benefits and favorable results; conversely, an individual will develop negative attitudes when considering that the results will be unfavorable. We consulted Ajzen [24], Yi and Gong [36] and define the attitudes toward value co-creation as an individual's opinion on value co-creation and his/her positive or negative feelings toward value co-creation behavior.

*1.4. Subjective Norms for Value Co-creation*

The concept of value co-creation refers to enterprises and consumers needing to interact with each other during a resource exchange process to exchange resources and create common values [9]. Consumers' active engagements in value co-creation can elevate the effectiveness and efficiency of the co-creation process; in addition, the values gained from co-creation can satisfy consumers and benefit enterprises [42]. Subjective norm is conceptualized as a person's perception of social norms regarding whether they should engage in a behavior or not [24]. The influences of subjective norms from consulted groups or individuals on an individual increase when social pressure or dependence willingness is heightened. Contrarily, the influences of subjective norms on an individual decline when social pressure or dependence willingness is lowered [43,44]. The present study consulted reference [36] and reference [24] and defines the subjective norms for value co-creation as the degree of identification held by an individual's significant others toward value co-creation.

## 2. Hypotheses Development and Research Model

Value co-creation refers to the values created through customer–enterprise collaboration to satisfy customer needs and benefit enterprises [12]. Attitude refers to an individual's positive or negative emotional reactions toward a certain target [45]. The relevant discourses on attitudes in previous studies have indicated that attitude is composed of three major elements: emotion, behavior, and cognition [46]. Customers holding positive attitudes toward value co-creation are subsequently more likely to adopt value co-creation behavior [47]. The aforementioned perspectives correspond with Ajzen [24] suggesting that behavioral intention determines a person's actual behavior and is influenced by attitude. Numerous previous studies have also verified the positive influences of attitude on behavior. Therefore, we propose the following hypothesis:

**Hypothesis 1(H1).** *Value co-creation attitudes positively influence value co-creation behavior.*

Subjective norm means that an individual's decision as to whether to engage in a behavior is influenced by his/her significant other's degree of identification toward a certain behavior [24]. A person with a strong subjective norm is more likely to engage in a certain behavior [36,48]. Psychological research indicates that the subjective norm is a critical decision-making factor associated with perceived usefulness [49] and behavioral intention [50], and that behavioral intention determines a person's actual behavior. The present study mainly investigates the subjective norms for value co-creation, which refers to the degree of identification toward value co-creation held by the significant others of an individual. A customer has a higher tendency to engage in value co-creation behavior when his/her significant other shows higher degrees of identification with the value co-creation offered by a fitness club. Numerous previous studies have also verified the positive influences of subjective norms on behavior. Therefore, we propose the following hypothesis:

**Hypothesis 2(H2).** *Value co-creation subjective norm positively influence value co-creation behavior.*

Price reasonability refers to an assessment as to whether a seller's price and buyer's price are reasonable or acceptable [51], and is a result derived from the reference price and actual paid price [52]. According to the perspective of equity theory proposed by Adams [53], people subjectively assess whether the costs they invested in were equivalent to the acquired rewards during an exchange relationship, thereby judging whether the transaction was fair. Consumers will consider a transaction as fair (unfair) when their perceived benefits corresponded (did not correspond) with the price they paid [54], and were less likely to engage in a behavior when they felt the transaction was unfair [51]. Regarding the consumer judgment of price reasonability, from the perspective of customer complaints, negative emotions or emotional reactions are aroused when a consumer considers the price at a fitness club to be unreasonable. Therefore, customer complaints are less likely to occur when customers consider the prices at fitness clubs to be reasonable and acceptable; moreover, customers will engage in co-creation behaviors such as feedback and assistance offering [55]. Contrarily, customers' complaints are likely to occur when customers consider the prices at fitness clubs to be unreasonable, thus hindering customers from engaging in value co-creation behavior. Consequently, we propose the following hypothesis:

**Hypothesis 3(H3).** *Price reasonability positively influences value co-creation behavior.*

Value co-creation indicates the high participation of customers and the production of highly-customized products and services developed based on customers' individual needs [35]. Prahalad and Ramaswamy [18] propose that customers are viewed as active value co-creators rather than passive recipients; therefore, enterprises must serve as promoters during the value co-creation process to satisfy customer needs [10]. Customer participation benefits both themselves and enterprises [8], and enables enterprises to develop relatively more intimate and profitable relationships with customers [17]. Customers will attempt to interact with enterprises through self-participation methods to acquire their expected services or products. Through customers' active participation, enterprises can lower the costs for enterprise–customer interactions and increase customer satisfaction [56].

Celsi and Olson [57] determined that customer participation in services influenced their own perceptions and loyalty, and is beneficial for developing customer loyalty at places designed for leisure and entertainment [58]. Baker, Cronin and Hopkins [59] investigated service participation and determined that customer participation increased customer loyalty for the leisure and service industries. Therefore, customers who engage in value co-creation behaviors at fitness clubs will become more familiar with the courses, environments, facilities, and other relevant information at the clubs because of the interaction and participation processes, thereby increasing their purchase intention and willingness to recommend the products or services to their acquaintances [60]. The following hypothesis is hence proposed:

**Hypothesis 4(H4).** *Value co-creation behavior positively influences loyalty.*

Pleshko and Al-Houti [61] report that a person with the following characteristics is viewed as a heavy user: a user who purchases more products than other people do, who has more product purchasing experiences than other people have, and who has more time to engage in purchases than other people have. Compared with light users, heavy users engage in purchases more frequently, possess more professional knowledge, have price sensitivity towards the products they care about, and actively collect relevant information [62]. Brunsø [63] determined that compared with light users, heavy users can more accurately assess product quality. Users gradually generate repeated purchase behavior as their knowledge and experiences increase [64,65]. The behavioral modes of consumers with regular behaviors are relatively more fixed, and consumers with relatively higher exercise intensity, and more time and frequency of participation are viewed as heavy users [66]. Customers of this type are aware that regular exercise can reduce their stress, improve their mood, and elevate the

effectiveness of their daily activities [67]; in addition, they are clearly aware of the time to exercise, the duration of exercise, and the exercise intensity they need to achieve, thereby attaining their perceived satisfaction at fitness clubs [62].

Therefore, when these types of consumers of regular behaviors participate in the of value co-creation, they will think that the participation of co-creation value is less helpful, leading to a lesser impact of co-creation value on loyalty. On the contrary, consumers without regular behaviors may be hampered by insufficient time to exercise and the effective intensity of each exercise, making the fitness club less familiar. Customers with such irregular behavior are less familiar with fitness clubs and thus need to engage in the following co-creation methods to successfully acquire the fitness club services that meet their needs: information seeking, knowledge sharing, personal interaction, message feedback, helping, and tolerance. Consequently, these customers consider value co-creation behavior to be more helpful and are more satisfied when they engage in value co-creation behavior, thus generating repeated purchase behavior [27] and considerably increasing their loyalty to the clubs. On the basis of the aforementioned statements, consumers with regular behaviors view participation in value co-creation behavior as associated with smaller benefits because they have higher exercise intensity and more time and frequency of exercise. However, consumers without regular behaviors consider value co-creation behavior as more helpful because it can improve the problem of lacking enough time for exercise, helping them to meet the required amount of exercise and achieve the exercise effects each time they are at the fitness clubs. Therefore, we propose the following hypothesis:

**Hypothesis 5(H5).** *Regular behaviors exhibit negative moderative effects between value co-creation behavior and loyalty.*

The research model is depicted in Figure 1.

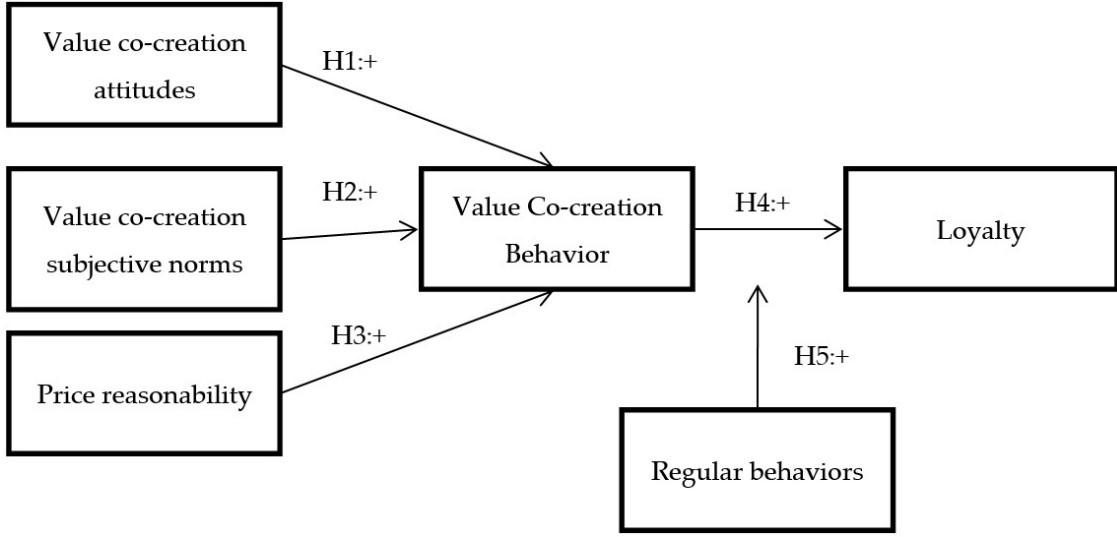

**Figure 1.** The research framework.

## 3. Methods

### 3.1. Research Instruments

The pre-test questionnaire in this study was based on the co-creation behavior scale compiled by Yi and Gong [36], Ajzen's [24] attitude and subjective norm scale, Voss's [68] price rationality scale, Lee's [31] loyalty scale, and according to regular behavior made by the Sports Administration Ministry of Education, judging by taking exercises at least 3 times a week, 30 minutes each time, and the intensity of the exercise would be sweating and wheezing. The subjects of this questionnaire are Taiwan fitness club users. In this study pre-test questionnaire, 150 questionnaires were issued, and 140 valid samples

were recovered with an effective recovery rate of 93.3%. SPSS 22.0 [69] statistical software was used to conduct the reliability and validity analysis of data archiving. A formal questionnaire was formed after the questions with poor reliability were deleted.

### 3.2. Reliability Analysis

This research executed the reliability analysis in SPSS 22.0 [69] and used the dimensional consistency between Cronbach's alpha examination and dimensional items. The results of all of Cronbach's alpha of the items were higher than 0.7, (Information seeking: 0.849, Information sharing: 0.874, Personal interaction: 0.868, Feedback: 0.823, Helping: 0.852, Tolerance: 0.802, Value co-creation attitudes: 0.847, Value co-creation attitudes subjective norms: 0.881, Price reasonability: 0.938, Loyalty: 0.806) indicating a high degree of internal consistency.

### 3.3. Validity Analysis

This study included convergent validity, construct reliability, and average variance extracted. In the measurement model, the standardized regression coefficient of each question and its corresponding variable were higher than 0.7. This showed that the scale had good convergence validity. The construct reliability was (Information seeking: 0.85, Information sharing: 0.87, Responsible behavior: 0.88, Personal interaction: 0.87, Feedback: 0.82, Helping: 0.85, Tolerance: 0.84, Attitudes toward value co-creation: 0.85, Subjective norms for value co-creation: 0.88, Price reasonability: 0.94, Loyalty: 0.81), and the Average Variance Extracted was (Information seeking: 0.66, Information sharing: 0.64, Responsible behavior: 0.64, Personal interaction: 0.57, Feedback: 0.61, Helping: 0.59, Tolerance: 0.58, Attitudes toward value co-creation: 0.65, Subjective norms for value co-creation: 0.65, Price reasonability: 0.84, Loyalty: 0.58). All that are stated above were in line with the inspection standard of reference [70].

After the valid questionnaires were returned, the statistical software SPSS was adopted for data archiving, and the software AMOS 22.0 [71] was adopted for offending estimates examination, normal distribution examination, confirmatory factor analysis, and structural relation analysis. The fitness club users in Taiwan were incorporated as the research participants, with a sampling error of no more than 4.5% and a confidence interval of 95%. Suhumacker and Lomax [72] indicate that SEM use a sample size the optimal condition is between 250 and 500. Therefore, the present study set the sample number as 450 and conducted a convenience sampling outside the fitness clubs in Taichung City from 20th May to 13th June 2016. A total of 470 questionnaires were distributed using convenience sampling, of which 453 usable samples were returned. The total response rate was 96.3%.

## 4. Results

### 4.1. Analysis of Sample Characteristics

This study recruited the fitness club users in Taichung City as the research samples. The formal questionnaire valid samples totaled 453 people in which 204 men and 249 women accounted for 45% and 55% of the valid samples, respectively. Participants with the following age group, marital status, and educational attainment accounted for the largest number among all the samples: 205 participants were aged 21–40 years old and accounted for 45.3% of the valid samples; 318 participants were married and accounted for 56.5% of the valid samples; 318 participants had a bachelor degree (junior college) and accounted for 70.2% of the valid samples. A total of 150 participants had regular behaviors, accounting for 33.1% of the valid samples and the percentage was similar 34.4%, which was the percentage of the population who exercise on a regular basis [73].

### 4.2. Test of Offending Estimates

Offending estimates indicates that the explanation of a structural mode or measurement model is inadequate if a statistical coefficient exceeds the acceptable range [74]. Therefore, we examined

whether offending estimates occurred before we conducted the overall goodness-of-fit test. The results revealed that the error variation of the estimated value was 0.03–0.08 and the standardized coefficient was 0.16–0.94, without exceeding the standard value of 0.95 and all conforming to the standard set by reference [74]. Consequently, offending estimates did not occur in the overall mode of this study, and the mode could thus be examined using the goodness of fit test.

### 4.3. Test of Normal Distribution

Considering that an expansion of the Chi square value might induce the research model to yield incorrect inference, this study followed the standard of normality test proposed by Kline [75] and judged that the bias of all the variables did not exceed the absolute value of 1, and that Kurtosis did not exceed 7. Therefore, this study accorded with the normal distribution for a single variable.

### 4.4. Linear Structural Relations Model Analysis

The study results revealed, CMIN/DF = 2.11, P = 0.000, GFI = 0.90, CFI = 0.94, AGFI = 0.83, RMSEA = 0.050, TLI = 0.94. It matches test standard suggested by [76,77]), which provided a good Linear Structural Relations Model.

(A) Path Analysis

Figure 2 displays the path analysis results of the relationships among all the constructs. The path value of the standardized coefficients of the attitudes toward value co-creation on value co-creation behavior was 0.54 ($p < 0.001$), with the results supporting H1, which indicated that fitness club users are influenced by their positive comments and positive feelings toward value co-creation behavior when they are engaged in value co-creation behavior. The path value of the standardized coefficients of the subjective norms for value co-creation was 0.34 ($p < 0.001$), with the results supporting H2, which suggests that fitness club users are likely to have higher engagement in value co-creation behavior when they perceive that the people and groups around them positively identify with the behavior. The path value of the standardized coefficients of price reasonability on value co-creation behavior was 0.09 ($p < 0.01$); thus, the results supported H3, indicating that positive influences are generated on co-creation behavior when consumers view the prices at fitness clubs as reasonable. The path value of the standardized coefficients of value co-creation behavior on loyalty was 0.87 ($p < 0.001$), with the results supporting H4, indicating that consumers are loyal to fitness clubs when they determine that participating in value co-creation behavior at the clubs can promote the production of highly-customized services to satisfy their needs. Therefore, H1, H2, H3, and H4 were all supported (see Table 1).

**Table 1.** Hypothesis testing.

| Hypotheses | Path | Estimate | S.E. | C.R. | *p* | Remarks |
|:---:|:---:|:---:|:---:|:---:|:---:|:---:|
| H1 | VCCA→VCCB | 0.484 | 0.086 | 5.607 | *** | accepted |
| H2 | VCCSN→VCCB | 0.338 | 0.096 | 3.517 | *** | accepted |
| H3 | Price Affordability→VCCB | 0.071 | 0.025 | 2.836 | 0.005 * | accepted |
| H4 | VCCB→Loyalty | 0.849 | 0.051 | 16.703 | *** | accepted |

\* $p < 0.05$, \*\*\* $p < 0.001$.

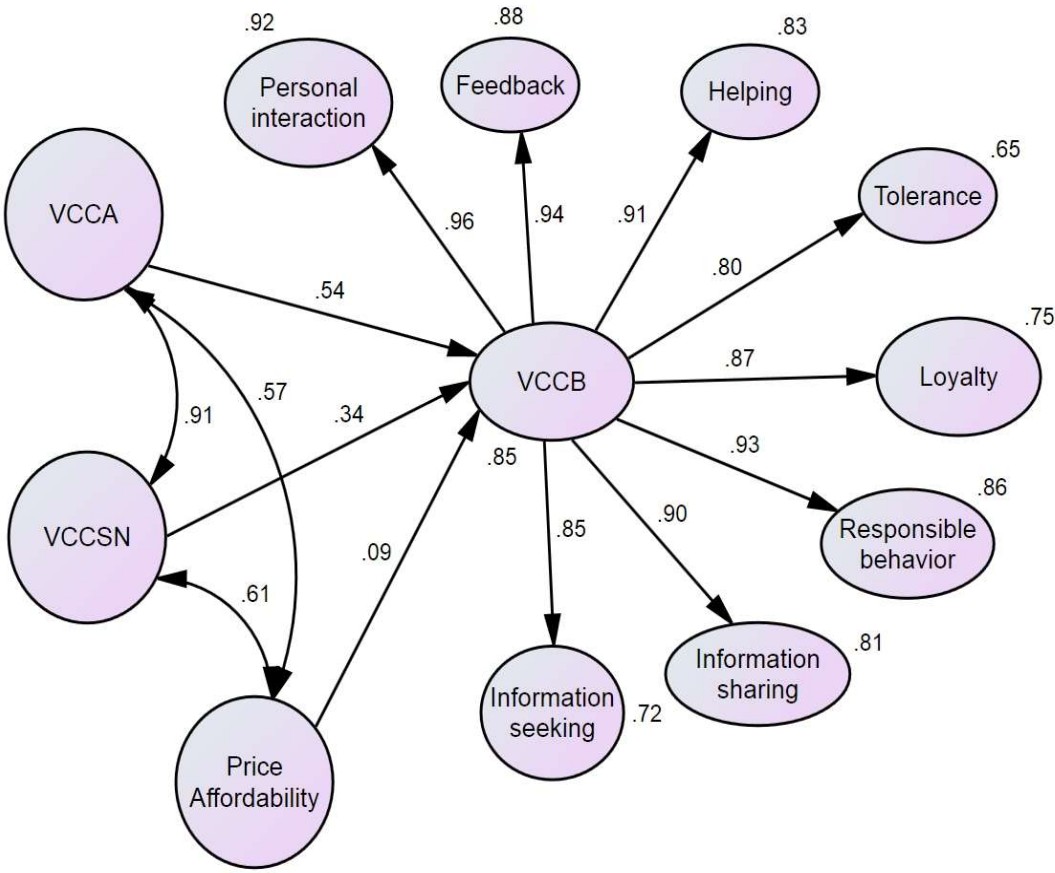

**Figure 2.** Path analysis of structure mode.

(B) Analysis of Regular Behavior Moderation

This study according to above-mentioned (regular behavior made by the Sports Administration Ministry of Education), we can separate the 453 samples into two groups which are regular behavior and irregular behavior to do inspect. It divided the subjects into two groups of regular behavior and irregular behavior for the test, and both groups met the sample number at least 5 times of the estimated parameter [78]. After the non-standardized coefficients in Table 2 were obtained, a Z value of −2.157 for regular behavior was obtained through the Fisher Z examination method, in which the absolute value was greater than the test standard of 1.96 [79]. The results indicated that the regular behavior in the present study yielded negative moderating effects [80], hence, H5 was supported.

**Table 2.** Regression analysis of regular behavior.

| Regular Behaviors | Path | Unstandardized Regression Coefficient | Standard Error | Standardized Regression Coefficient |
|---|---|---|---|---|
| regular behavior(N=150) | value co-creation behavior → loyalty | 0.68 *** | 0.086 | 0.72 *** |
| without regular behavior.(N=303) | value co-creation behavior → loyalty | 0.91 *** | 0.063 | 0.90 *** |

*** $p < 0.001$.

## 5. Discussion and Conclusions

This study mainly explored the relationships among Taiwan fitness club users' attitudes toward value co-creation, subjective norms for value co-creation, price reasonability, value co-creation behavior,

and loyalty. We also examined whether regular behavior generated moderating effects between value co-creation behavior and loyalty.

## 5.1. General discussion

The verified results revealed consumers were influenced by their positive opinions or attitudes toward value co-creation behavior during their participation in the behavior, thereby heightening their tendency to engage in the value co-creation behavior. The subjective norms for value co-creation positively influence value co-creation behavior. Consumers developed positive value co-creation behavior during their participation in the behavior when they perceived that their significant others positively identified with the behavior. Consumers were willing to engage in value co-creation behavior when they felt that the price at fitness clubs was reasonable and acceptable. Customers participation benefitted both customers and enterprises, and in turn positively affected customers' loyalty to fitness clubs. The results offered a type of new method for increasing customer loyalty to compensate for the previous research on building customer loyalty in the field of leisure industry.

The results of this study show that consumers with irregular behavior have had a more positive effect on loyalty than those with regular behavior. The reason is that the pattern of consumer behavior with regular behavior is relatively fixed, having more participation time and higher exercise intensity. This type of consumer engaged with more time and frequency in value co-creation behavior and demonstrated higher exercise intensity, and they knew what time and how long they should exercise, and the exercise intensity index they need to attain. Therefore, this type of consumer believes that the help they gained from value co-creation was smaller during their engagement in value co-creation behavior; thus, the influences of their value co-creation behavior on loyalty were weaker. The consumers without regular behavior lacked enough time for exercise and thus exercised less than three times per week, exercised less than 30 minutes per time, or did not meet the effective exercise intensity each time. Under this condition of irregular behavior, consumers were less familiar with fitness clubs and thus required the following methods to complete their exercise: information seeking, knowledge sharing, personal interaction, feedback, helping, and tolerance. Hence, these consumers believed that engaging in value co-creation behavior was more helpful and satisfying, thereby generating repeated purchase behavior and considerably elevating their loyalty to fitness clubs.

## 5.2. Management Implications

Value co-creation indicates that consumers and businesses collaborate with each other to create services that satisfy the needs of both parties [12]. In the 21st century, fitness clubs need to maintain good interactions with consumers and enhance consumers' relevant knowledge on value co-creation. The goals are to encourage consumers to understand the benefits of value co-creation, to interact with the clubs, and to specifically indicate their needs, thereby increasing their loyalty to the fitness clubs and enabling club management personnel to implement correct management actions. The competitiveness of value co-creation can be elevated if consumers can propose concrete suggestions for the clubs from their own perspectives and offer correct information to clubs' relevant personnel to improve the possible issues encountered by the clubs.

All consumers have different attitudes because of their different emotions and cognitions [46]. Consumers who hold positive attitudes toward value co-creation are naturally more likely to adopt the value co-creation behavior. Therefore, fitness clubs need to offer persuasive messages, or to collaborate with persuasive spokespersons to illustrate the benefit of value co-creation, encouraging consumers to understand and participate that engaging in value co-creation behavior is worthwhile because it is joyful, beneficial, and imperative. Users' decisions as whether to engage in a behavior is also influenced by the degree of identification of their significant others [24]; that is, individuals are more likely to engage in value co-creation behavior if their significant others highly support value co-creation behavior. Therefore, fitness clubs' management personnel can offer professional instructions and hold

various exchange activities to encourage consumers and their family or friends to join the clubs and increase their support for value co-creation.

A company's products and price will influence consumers' perceived fairness [81]. Consumers will take the other companies that sell same products into consideration when judging price fairness [82]. Hence, we recommend that corporate personnel use Internet resource platforms to promote the relevant mechanisms at their fitness clubs (e.g., relevant information on charging standards or equipment usage guide) and deliver the benefits to consumers through visual images, titles, or explanations from professional personnel. These methods enable consumers to gain an in-depth understanding of, and identification with, the value co-creation behavior and sincerely view the acquired results as reasonable, thereby increasing their willingness to purchase the products [82].

Consumers without regular behavior are viewed as light users who want to purchase perfect products; however, these consumers lack professional knowledge on how to judge products and usually judge products' quality by looking at them [63]. Compared with heavy users, light users' loyalty to fitness clubs are more influenced by value co-creation behavior. Therefore, we recommend that fitness club management personnel pay extra attention to which type of consumers visit the clubs, and encourage light users to engage in value co-creation behavior when these users visit the clubs, thereby considerably increasing consumer loyalty to the clubs.

### 5.3. Research Contributions

This study verified the scale of value co-creation behavior explored by Yi and Gong [36]; strengthened the research of Cossío-Silva [19] on verifying the influences of customer perspective on their attitudinal and behavioral loyalty; investigated the influences of antecedent and moderating variables on value co-creation behavior; and added the completeness of the application of the theory of value co-creation on the leisure industry.

We determined that customers are more willing to engage in value co-creation behavior when they hold positive attitudes toward value co-creation, when their significant others positively identified with value co-creation, and when they considered the selling price and the obtained benefits and values to be acceptable. This study was employed to strengthen the concept of co-creation role exploration previously conducted by Bendapudi and Leone, Prahalad and Ramaswamy [17,18] enabling enterprises and customers to develop and build favorable interactive relationships and simultaneously elevate companies' overall operational performance. Moreover, this study determined that regular behavior exhibits negative moderating effects on value co-creation behavior and loyalty, indicating that value co-creation can effectively elevate light users' loyalty toward brands.

Relevant previous studies on loyalty in the leisure industry mostly focus on variables such as service quality and perceived values; however, the value co-creation mentioned in the manufacturing industry has been sparse in the literature on the sports and leisure industry. The present study incorporated the theory of value co-creation into the leisure and service industry, and the results verified that value co-creation behavior can effectively elevate customer loyalty, providing a different research direction for the leisure and sports domain that emphasizes interactive experiences and services.

### 5.4. Limitations and Suggestions

This study merely explored consumers' regular behaviors; however, numerous factors affect consumer attitude and loyalty. For example, the number of people that accompany a consumer may generate different degrees of identification toward value co-creation behavior, and thus may have different influences on the consumer. Consumers who visit fitness clubs alone may more easily identify with the value co-creation behavior through persuasive communication methods and in turn develop consumer behavior. Consumers accompanied by others may be more easily influenced by external factors; therefore, the attempts to arouse consumers' curiosity and motivation can possibly guide them

to satisfy their personal needs [83]. Subsequent studies may want to incorporate value co-creation behavior as an antecedent variable.

Moreover, the sports and service industries are very common in Taiwan; however, this study merely recruited the users of privately-owned fitness clubs as the research participants. Civil sports centers are subsidized and constructed by the Sports Administration of the Ministry of Education and are then entrusted to non-governmental organizations for management. The hardware facilities, number of service personnel, and professionalism at the civil sports centers may differ considerably from the private fitness clubs, and the degree of value co-creation behavior offered at civil sports centers and private fitness clubs may also differ greatly. Hence, we recommend that future studies incorporate the consumers at the sports centers subsidized and built by the government and managed by non-governmental organizations as the research participants for comparative analysis, thereby determining whether the value co-creation behavior at different types of sports centers exhibit different effects and influences on customer loyalty. In addition, the influences of different types of service industry on value co-creation behavior may also differ. For example, the influences of value co-creation behavior may be smaller for the catering industry because the duration of consumer participation at restaurants is shorter, but may be larger for the tourism industry because the duration of consumer participation is longer. Future studies can further explore this topic.

**Author Contributions:** Conceptualization and writing, Y.-L.L. and L.-Y.P.; Methodology, C.-H.H.; Formal Analysis, D.-C.L.

**Funding:** This research received no external funding.

**Conflicts of Interest:** The authors declare no conflict of interest.

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
