# Peer review of "Exploring the Sustainability Correlation of Value Co-Creation and Customer Loyalty-A Case Study of Fitness Clubs"

_sustainability, doi:10.3390/su11010097_

Round 1
Reviewer 1 Report
There are some sections within the Research Methods section that might be better presented within the Introduction. I was unclear on how many populations were studied. I detected between one and three separate populations. I believe that this study could be beneficial to operators of fitness clubs. However, I was unclear on the management implications; specifically, whether the main goal of value co-creation is to benefit the health outcome of patrons by increasing service consumption, or to benefit the health of the business through paid service consumption.
Specific line comments below:
Line 31. Citation needed for first sentence
Line 33. Change attract(s).
Line 36. Does a service company provide goods?
Line 38. In what ways to the parties interact? Is this a negotiation?
Line 38, 39. I do not understand citation 8.
Line 40. What is high service? Is this customer service? Is this the number of services?
Line 43, 44, 45. I do not understand these two sentences.
Line 51. Change lie(s).
Line 57. Remove “a” in front of research.
Line 58. Remove “the” prior to co-creation.
Line 59. Add “value” prior to co-creation
Line 60. Is the expenditure by the customer?
Line 60. Value should not be capitalized. Numerous instances throughout. Additionally, co-creation should not be capitalized.
Line 60. Is satisfaction and expenditures by a customer loyalty?
Line 61. Remove “es” from researches.
Line 68. From should not be capitalized.
Line 73. Should you define or cite value co-creation theory and planned behavior theory here?
Line 76. Which price are you referencing?
Line 80. Behaviors such as what?
Line 100. There is a “c” and an “o” that need to be fixed.
Line 104. This section should be part of the introduction.
Line 114. This section should be part of the introduction and might be best to be included early in the introduction to clarify the meaning of value co-creation.
Line 129. Should “consumers” be changed to “producers”?
Line 151. Names of those consulted for citation 35 and 24 should be included.
Line 159, 160, 161. I do not understand the sentence for citation 24.
Line 174. “Suggestion” should be “suggesting”.
Line 208. What is spontaneous behavior?
Line 209. Why would companies not need to offer successful products or services?
Line 237. Can you cite the 80/20 rule?
Line 253. Which type of consumer?
Line 253. Add “of” prior to value.
Line 255. What is irregular behavior?
Line 289. A name should be added for citation 35.
Line 289. Should be Ajzen’s.
Line 291. I don’t understand how the questionnaire is “judged”.
Line 293. Where did the pre-test questionnaire come from?
Line 296. Please cite SPSS.
Line 304. Should be Cronbach’s Alpha. Also, can we state which constructs are above a minimum .70 Cronbach’s Alpha. Perhaps within a chart to include the correlation coefficients?
Line 305. Correlation is between two numbers not around them.
Line 324, 325. Should all of these words be capitalized? Do they refer to constructs?
Line 327. Construct should not be capitalized.
Line 337. Cite AMOS software.
Line 341. I don’t understand the use of scanning electron microscopy.
Line 342. How was this randomized. It seems as though a convenience sample was chosen.
Line 350. Are these 435 individuals part of a separate study?
Line 375, 376. It is hard to read this presentation of the numbers.
Line 378. Could the path analysis results be better presented in a table?
Line 401. Does 7,333 refer to individuals? If so, is this a separate study?
Line 417. Remove the last sentence.
Line 421. Can you define the behavior?
Line 487. Include name prior to citation 35 and 19.
Line 495. Please define “scale”.
Line 496. Include name prior to citation 17 and 18.
Line 518. Please provide citation for sentence ending with “behavior”.
Author Response
Response to Reviewer 1 Comments
There are some sections within the Research Methods section that might be better presented within the Introduction.
Thank reviewer for your suggestions and reminders. We modified the text locally to : additional remarks from line 103 to line 108. Additional contents:(This study surveys Taiwan’s fitness club users, taking Taiwan fitness club as an example, exploring the study on the relationships between value co-creation behavior, the concept of value co-creation attitude, the subjective norm of value co-creation, price rationality and loyalty. In addition, the study also includes regular behaviors to test whether there will be a moderating effect on co-value behavior and loyalty. This study development of the questionnaire began with a review of the relevant literature.) (In blue)
I was unclear on how many populations were studied. I detected between one and three separate populations.
Thank reviewer for your suggestions and reminders. We modified the text locally to : additional remarks from line 294 to line 297. Additional contents:(The subjects of this questionnaire are Taiwan fitness club users. In this study pre-test questionnaire, 150 questionnaires were issued, and 140 valid samples were recovered with an effective recovery rate of 93.3%. ) (In blue)
Thank reviewer for your suggestions and reminders. We modified the text locally to : additional remarks from line 325 to line 329. Additional contents:(Suhumacker and Lomax [72] indicate that SEM uses a sample size the optimal condition is between 250 and 500. Therefore, the present study set the sample number as 450 and conducted a convenience sampling outside the fitness clubs in Taichung City from May 20th to June 13th , 2016. A total of 470 questionnaires were distributed using convenience sampling, of which 453 usable samples were returned. The total response rate was 96.3%.) (In blue)
Thank reviewer for your suggestions and reminders. We modified the text locally to : additional remarks from line 391 to line 393. Additional contents:(This study according to above-mentioned (regular behavior made by the Sports Administration Ministry of Education), we can separate the 453 samples into two groups which are regular behavior and irregular behavior to do inspect.) (In blue)
I believe that this study could be beneficial to operators of fitness clubs. However, I was unclear on the management implications; specifically, whether the main goal of value co-creation is to benefit the health outcome of patrons by increasing service consumption, or to benefit the health of the business through paid service consumption.
Thank reviewer for your suggestions and reminders. We modified the text locally to : additional remarks from line 440 to line 441. Additional contents:(Value co-creation indicates that consumers and businesses collaborate with each other to create services that satisfy the needs of both parties [12].) (In blue)
Additional remarks from line 452 to line 456. Additional contents:(Therefore, fitness clubs need to offer persuasive messages, or to collaborate with persuasive spokespersons to illustrate the benefit of value co-creation, encouraging consumers to understand and participate that engaging in value co-creation behavior is worthwhile because it is joyful, beneficial, and imperative.) (In blue)
Additional remarks from line 459 to line 461. Additional contents:(Therefore, fitness clubs’ management personnel can offer professional instructions and hold various exchange activities to encourage consumers and their family or friends to join the clubs and support for value co-creation.) (In blue)
Line 31. Citation needed for first sentence
Thank reviewer for your suggestions and reminders. We modified the text locally to : additional remarks from line 35 to line 37. Additional contents:(IHRSA [5] there are more than 180,000 fitness clubs around the world with an output value of more than $8.4 million. Whether counting by membership or by users, their numbers are growing year by year.) (In blue)
Line 33. Change attract(s).
Thank reviewer for your suggestions and reminders. We modified the text locally. (In blue)
Line 36. Does a service company provide goods?
Thank reviewer for your suggestions and reminders. We modified the text locally to : additional remarks from line 39 to line 40. Additional contents:(When customers have a high degree of satisfaction from the service or product of the fitness club provide) (In blue)
Line 38. In what ways to the parties interact? Is this a negotiation? Thank reviewer for your suggestions and reminders. We modified the text locally to : additional remarks from line 43 to line 44. Additional contents:(There are few opportunities for both parties to interact to get consumers feedbacks [7]) (In blue)
Line 38, 39. I do not understand citation 8. Thank reviewer for your suggestions and reminders. We modified the text locally to : additional remarks from line 44 to line 45. Additional contents:(so that they easy to neglect the benefits from interact of companies and customers [8].) (In blue)
Line 40. What is high service? Is this customer service? Is this the number of services?
Thank reviewer for your suggestions and reminders. We modified the text locally to : additional remarks from line 45 to line 46. Additional contents:(With fierce competition and diversified choices, customers have more requirements for products quality and services quality.) (In blue)
Line 43, 44, 45. I do not understand these two sentences.
Thank reviewer for your suggestions and reminders. We modified the text locally to : additional remarks from line 48 to line 51. Additional contents:(However, in this process, enterprises and customers need to interact and exchange resources to create a common value [9]. But customers and businesses need to maintain mutual aid and dependence to create additional value and reasonable distribution, and improved satisfaction [10].) (In blue)
Line 51. Change lie(s).
Thank reviewer for your suggestions and reminders. We modified the text locally. (In blue)
Line 57. Remove “a” in front of research.
Thank reviewer for your suggestions and reminders. We modified the text locally. (In blue)
Line 58. Remove “the” prior to co-creation.
Thank reviewer for your suggestions and reminders. We modified the text locally. (In blue)
Line 59. Add “value” prior to co-creation
Thank reviewer for your suggestions and reminders. We modified the text locally. (In blue)
Line 60. Is the expenditure by the customer?
Thank reviewer for your suggestions and reminders. We modified the text locally to : additional remarks from line 64 to line 65. Additional contents:(Grissemann [20] also found that participation in co-creation has a positive impact on customer satisfaction.) (In blue)
Line 60. Value should not be capitalized. Numerous instances throughout. Additionally, co-creation should not be capitalized.
Thank reviewer for your suggestions and reminders. We modified the text locally. (In blue)
Line 60. Is satisfaction and expenditures by a customer loyalty?
Thank reviewer for your suggestions and reminders. We modified the text locally. (In blue)
Line 61. Remove “es” from researches.
Thank reviewer for your suggestions and reminders. We modified the text locally. (In blue)
Line 68. From should not be capitalized.
Thank reviewer for your suggestions and reminders. We modified the text locally. (In blue)
Line 73. Should you define or cite value co-creation theory and planned behavior theory here?
Thank reviewer for your suggestions and reminders. We modified the text locally to : additional remarks from line 78 to line 80. Additional contents:(Therefore, verify whether the co-creation attitude and subjective norm of value co-creation are the antecedent variable for co-creation behavior. This is the third objective of this study.) (In blue)
Line 76. Which price are you referencing?
Thank reviewer for your suggestions and reminders. We modified the text locally to : additional remarks from line 81. Additional contents:(In addition, the price is an important antecedent variable for decision behavior [25].) (In blue)
Line 80. Behaviors such as what?
Thank reviewer for your suggestions and reminders. We modified the text locally. (In blue)
Line 100. There is a “c” and an “o” that need to be fixed.
Thank reviewer for your suggestions and reminders. We modified the text locally. (In blue)
Line 104. This section should be part of the introduction.
Thank reviewer for your suggestions and reminders. We modified the text locally. (In blue)
Line 114. This section should be part of the introduction and might be best to be included early in the introduction to clarify the meaning of value co-creation.
Thank reviewer for your suggestions and reminders. We modified the text locally. (In blue)
Line 129. Should “consumers” be changed to “producers”?
Thank reviewer for your suggestions and reminders. We modified the text locally to : additional remarks from line 132 to line 134. Additional contents:(Because of those consumers participate, let companies have able to produce highly-customized products and services, which are created based on consumer needs [35].) (In blue)
Line 151. Names of those consulted for citation 35 and 24 should be included.
Thank reviewer for your suggestions and reminders. We modified the text locally. (In blue)
Line 159, 160, 161. I do not understand the sentence for citation 24. Thank reviewer for your suggestions and reminders. We modified the text locally to : additional remarks from line 163 to line 165. Additional contents:(Subjective norm is conceptualized as a person’s perception of social normative and family that they should engage in a behavior or not [24]. (In blue)
Line 174. “Suggestion” should be “suggesting”.
Thank reviewer for your suggestions and reminders. We modified the text locally. (In blue)
Line 208. What is spontaneous behavior?
Thank reviewer for your suggestions and reminders. We modified the text locally to : additional remarks from line 207 to line 210. Additional contents:(Therefore, customer complaints are less likely to occur when customers consider the prices at fitness clubs to be reasonable and acceptable; moreover, customers will engage in co-creation behaviors such as feedback and assistance offering [55].) (In blue)
Line 209. Why would companies not need to offer successful products or services?
Thank reviewer for your suggestions and reminders. We modified the text locally to : additional remarks from line 216 to line 218. Additional contents:(Value co-creation indicates the high participation of customers and the production of highly-customized products and services developed based on customers’ individual needs [35]. (In blue)
Line 237. Can you cite the 80/20 rule?
Thank reviewer for your suggestions and reminders. We modified the text locally. (In blue)
Line 253. Which type of consumer?
Thank reviewer for your suggestions and reminders. We modified the text locally to : additional remarks from line 253 to line 255. Additional contents:(Therefore, when these types of consumers of Regular behaviors participate in the of value co-creation, they will think that the participation of co-creation value is less helpful, leading to a lesser impact of co-creation value on loyalty. (In blue)
Line 253. Add “of” prior to value.
Thank reviewer for your suggestions and reminders. We modified the text locally. (In blue)
Line 255. What is irregular behavior?
Thank reviewer for your suggestions and reminders. We modified the text locally to : additional remarks from line 255 to line 257. Additional contents:(On the contrary, consumers with without regular behavior may be hampered by insufficient time to exercise and the effective intensity of each exercise, making the fitness club less familiar.) (In blue)
Line 289. A name should be added for citation 35.
Thank reviewer for your suggestions and reminders. We modified the text locally. (In blue)
Line 289. Should be Ajzen’s.
Thank reviewer for your suggestions and reminders. We modified the text locally. (In blue)
Line 291. I donx’t understand how the questionnaire is “judged”.
Thank reviewer for your suggestions and reminders. We modified the text locally to : additional remarks from line 291 to line 294. Additional contents:(and according to regular behavior made by the Sports Administration Ministry of Education, judging by taking exercises at least 3 times a week, 30 minutes each time, and the intensity of the exercise would be sweating and wheezing.) (In blue)
Line 293. Where did the pre-test questionnaire come from?
Thank reviewer for your suggestions and reminders. We modified the text locally to : additional remarks from line 289 to line 292. Additional contents:(The pre-test questionnaire in this study was based on the co-creation behavior scale compiled by Yi and Gong [36], Ajzen’s [24] attitude and subjective norm scale, Voss’s [68] price rationality scale, Lee’s [31] loyalty scale, and according to regular behavior made by the Sports Administration Ministry of Education) (In blue)
Line 296. Please cite SPSS.
Thank reviewer for your suggestions and reminders. We modified the text locally. (In blue)
Line 304. Should be Cronbach’s Alpha. Also, can we state which constructs are above a minimum .70 Cronbach’s Alpha. Perhaps within a chart to include the correlation coefficients?
Thank reviewer for your suggestions and reminders. When we measure reliability the SPSS providing two index that can decide the questionnaire’s items delete or not: one is correlation coefficient after deleting the item another one is Cronbach’s Alpha value after deleting the item. Due to Cronbach’s Alpha value after deleting the item doesn’t increasing, thus, the study list first index only, to avoid other researcher reading the article to generate misunderstanding. We modified the text locally to : additional remarks from line 300 to line 306. Additional contents:(This research executed the reliability analysis in SPSS 22.0 [69] and used the dimensional consistency between Cronbach’s alpha examination and dimensional items. The results the all of Cronbach’s alpha of the items were higher than 0.7, (Information seeking:0.849、Information sharing:0.874、Personal interaction:0.868、Feedback:0.823、Helping:0.852、Tolerance:0.802、Value co-creation attitudes:0.847、Value co-creation attitudes subjective norms:0.881、Price reasonability:0.938、Loyalty:0.806) indicating a high degree of internal consistency.) (In blue)
Line 305. Correlation is between two numbers not around them.
Thank reviewer for your suggestions and reminders. We modified the text locally. (In blue)
Line 324, 325. Should all of these words be capitalized? Do they refer to constructs?
Thank reviewer for your suggestions and reminders. We modified the text locally to : additional remarks from line 308 to line 309. Additional contents:(This study included convergent validity, construct reliability, and average variance extracted. (In blue)
Line 327. Construct should not be capitalized.
Thank reviewer for your suggestions and reminders. We modified the text locally to : additional remarks from line 308 to line 309. Additional contents:(This study included convergent validity, construct reliability, and average variance extracted.) (In blue)
Line 337. Cite AMOS software.
Thank reviewer for your suggestions and reminders. We modified the text locally. (In blue)
Line 341. I don’t understand the use of scanning electron microscopy.
Thank reviewer for your suggestions and reminders. We modified the text locally to : additional remarks from line 325 to line 326. Additional contents:(Suhumacker and Lomax [72] indicate that SEM use a sample size the optimal condition is between 250 and 500.) (In blue)
Line 342. How was this randomized. It seems as though a convenience sample was chosen.
Thank reviewer for your suggestions and reminders. We modified the text locally to : additional remarks from line 326 to line 328. Additional contents:(Therefore, the present study set the sample number as 450 and conducted a convenience sampling outside the fitness clubs in Taichung City from May 20th to June 13th , 2016.) (In blue)
Line 350. Are these 435 individuals part of a separate study?
Thank reviewer for your suggestions and reminders. We modified the text locally to : additional remarks from line 328 to line 329. Additional contents:(A total of 470 questionnaires were distributed using convenience sampling, of which 453 usable samples were returned. The total response rate was 96.3%.) (In blue)
Line 375, 376. It is hard to read this presentation of the numbers.
Thank reviewer for your suggestions and reminders. The study results revealed see table 2.
Table 2 Test of Normal Distribution
Variable | min | max | skew | c.r. | kurtosis | c.r. |
h4 | 1.000 | 7.000 | -.569 | -4.946 | -.265 | -1.149 |
pi5 | 1.000 | 7.000 | -.763 | -6.631 | .194 | .845 |
pi4 | 1.000 | 7.000 | -.702 | -6.099 | -.109 | -.472 |
rb1 | 1.000 | 7.000 | -.592 | -5.147 | -.112 | -.486 |
sh4 | 1.000 | 7.000 | -.326 | -2.836 | -.509 | -2.209 |
SNVCC4 | 2.000 | 7.000 | -.481 | -4.177 | -.269 | -1.167 |
PA1 | 1.000 | 7.000 | -.435 | -3.777 | .048 | .210 |
PA2 | 1.000 | 7.000 | -.467 | -4.055 | -.167 | -.727 |
PA3 | 1.000 | 7.000 | -.623 | -5.416 | .087 | .376 |
SNVCC1 | 1.000 | 7.000 | -.566 | -4.915 | -.041 | -.179 |
SNVCC2 | 1.000 | 7.000 | -.573 | -4.980 | -.162 | -.706 |
SNVCC3 | 1.000 | 7.000 | -.463 | -4.025 | -.251 | -1.091 |
AVCC1 | 2.000 | 7.000 | -.487 | -4.231 | -.442 | -1.921 |
AVCC2 | 1.000 | 7.000 | -.608 | -5.279 | .103 | .446 |
AVCC3 | 1.000 | 8.000 | -.596 | -5.178 | .093 | .405 |
pi1 | 1.000 | 7.000 | -.560 | -4.865 | .001 | .004 |
pi2 | 2.000 | 7.000 | -.262 | -2.272 | -.487 | -2.116 |
pi3 | 1.000 | 7.000 | -.478 | -4.153 | -.215 | -.933 |
fb1 | 1.000 | 7.000 | -.474 | -4.118 | -.129 | -.563 |
fb2 | 1.000 | 7.000 | -.590 | -5.129 | .181 | .787 |
fb3 | 1.000 | 7.000 | -.520 | -4.521 | -.174 | -.754 |
h1 | 1.000 | 7.000 | -.595 | -5.168 | -.013 | -.058 |
h2 | 1.000 | 7.000 | -.757 | -6.575 | -.053 | -.230 |
h3 | 1.000 | 8.000 | -.498 | -4.328 | .130 | .563 |
t1 | 1.000 | 7.000 | -.487 | -4.234 | -.255 | -1.106 |
t2 | 1.000 | 7.000 | -.580 | -5.038 | .283 | 1.229 |
t3 | 1.000 | 7.000 | -.622 | -5.400 | .162 | .704 |
sk3 | 1.000 | 7.000 | -.456 | -3.961 | -.421 | -1.829 |
sk2 | 1.000 | 7.000 | -.334 | -2.903 | -.232 | -1.009 |
sk1 | 1.000 | 7.000 | -.397 | -3.447 | -.544 | -2.365 |
sh3 | 1.000 | 7.000 | -.642 | -5.577 | .174 | .755 |
sh2 | 1.000 | 7.000 | -.383 | -3.328 | -.588 | -2.553 |
sh1 | 1.000 | 7.000 | -.392 | -3.404 | -.233 | -1.011 |
rb2 | 1.000 | 7.000 | -.533 | -4.634 | -.028 | -.122 |
rb3 | 1.000 | 7.000 | -.490 | -4.258 | -.272 | -1.182 |
rb4 | 1.000 | 7.000 | -.673 | -5.850 | .069 | .299 |
Loyalty3 | 1.000 | 7.000 | -.570 | -4.950 | -.015 | -.067 |
Loyalty2 | 1.000 | 7.000 | -.366 | -3.180 | -.300 | -1.302 |
Loyalty1 | 1.000 | 7.000 | -.567 | -4.925 | .228 | .990 |
Multivariate | 303.590 | 57.130 |
Line 378. Could the path analysis results be better presented in a table?
Thank reviewer for your suggestions and reminders. We modified the text locally to : additional remarks from line 380 to line 381. Additional contents:(
Table 1 Hypothesis testing
Hypotheses | Path | Estimate | S.E. | C.R. | p | Remarks |
H1 | VCCAàVCCB | 0.484 | 0.086 | 5.607 | *** | accepted |
H2 | VCCSNàVCCB | 0.338 | 0.096 | 3.517 | *** | accepted |
H3 | Price Affordability àVCCB | 0.071 | 0.025 | 2.836 | 0.005 | accepted |
H4 | VCCBàLoyalty | 0.849 | 0.051 | 16.703 | *** | accepted |
) (In blue)
Line 401. Does 7,333 refer to individuals? If so, is this a separate study?
Thank reviewer for your suggestions and reminders. We modified the text locally to : additional remarks from line 386 to line 388. Additional contents:(This study according to above-mentioned (regular behavior made by the Sports Administration Ministry of Education), we can separate the 453 samples into two groups which are regular behavior and irregular behavior to do inspect.)
According to the Sports Administration Ministry of Education, judging by taking exercises at least 3 times a week, 30 minutes each time, and the intensity of the exercise would be sweating and wheezing. if you out of this rule is mean without regular behavior. (In blue)
Line 417. Remove the last sentence.
Thank reviewer for your suggestions and reminders. We modified the text locally. (In blue)
Line 421. Can you define the behavior?
Thank reviewer for your suggestions and reminders. We modified the text locally to : additional remarks from line 405 to line 407. Additional contents:(The verified results revealed consumers were influenced by their positive opinions or attitudes toward value co-creation behavior during their participation in the behavior, thereby heightening their tendency to engage in the value co-creation behavior. )(In blue)
Line 487. Include name prior to citation 35 and 19.
Thank reviewer for your suggestions and reminders. We modified the text locally. (In blue)
Line 495. Please define “scale”
Thank reviewer for your suggestions and reminders. We modified the text locally to : additional remarks from line 475 to line 476. Additional contents:(This study verified the scale of value co-creation behavior explored by Yi and Gong [36]; strengthened the research of Cossío-Silva [19] (In blue)
Line 496. Include name prior to citation 17 and 18.
Thank reviewer for your suggestions and reminders. We modified the text locally. (In blue)
Line 518. Please provide citation for sentence ending with “behavior”.
Thank reviewer for your suggestions and reminders. We modified the text locally. (In blue)

Reviewer 2 Report
Dear author/s,
thank you for an interesting article. I consider your study as significant and useful for the field researched.
I have following remarks to increase the quality of the paper:
Revise the introduction part.
The paper aim and objective should presented better.
The chapter titles are confusing and improper (please, check it)
When you write about other studies you should support them by the literature reference (it is missing for example in the l. 26, 62, or 176)
Your stylistics and the language nature is sometimes too complicated (please revise it in the whole paper, is well as revise your hypotheses, again).
Figure 2 is too complicated and I am not sure it is so necessary to be a part of your paper. I´d recommend to find a better illustration.
Author Response
Response to Reviewer 2 Comments
Thank you for an interesting article. I consider your study as significant and useful for the field researched.
I have following remarks to increase the quality of the paper:
Revise the introduction part.
The paper aim and objective should presented better.
Thank reviewer for your suggestions and reminders. We modified the text locally to : additional remarks from line 103 to line 108. Additional contents:(This study surveys Taiwan’s fitness club users, taking Taiwan fitness club as an example, exploring the study on the relationships between value co-creation behavior, the concept of value co-creation attitude, the subjective norm of value co-creation, price rationality and loyalty. In addition, the study also includes regular behaviors to test whether there will be a moderating effect on co-value behavior and loyalty. This study development of the questionnaire began with a review of the relevant literature.) (In blue)
The chapter titles are confusing and improper (please, check it)
Thank reviewer for your suggestions and reminders. We modified the text locally. (In blue)
When you write about other studies you should support them by the literature reference (it is missing for example in the l. 26, 62, or 176)
Thank reviewer for your suggestions and reminders. We modified the text locally. (In blue)
Your stylistics and the language nature is sometimes too complicated (please revise it in the whole paper, is well as revise your hypotheses, again).
Thank reviewer for your suggestions and reminders. We modified the text locally. (In blue)
Figure 2 is too complicated and I am not sure it is so necessary to be a part of your paper. I´d recommend to find a better illustration.
Thank reviewer for your suggestions and reminders. We modified the text locally to : additional remarks from line 381. (In blue)

Reviewer 3 Report
In the assessment of the paper submitted for the review, I specifically focused on the discussed issues, applied methodology, the substantive content of the paper and its structure.
The subject area discussed in the paper should be considered topical.
The considerations conducted in the paper are focused on such categories as: price affordability, value co-creation behaviour, customer loyalty, regular behaviour.
The structure of the paper is clear. The value of the paper results from appropriate combination of literature studies with the results of an empirical research.
The reviewed paper has a lot of values and has a scientific nature.
In the abstract of the article, information about the purpose and about the applied method should be included, and the period in which the study was conducted ought to be indicated.
Author Response
Response to Reviewer 3 Comments
In the assessment of the paper submitted for the review, I specifically focused on the discussed issues, applied methodology, the substantive content of the paper and its structure.
The subject area discussed in the paper should be considered topical.
The considerations conducted in the paper are focused on such categories as: price affordability, value co-creation behaviour, customer loyalty, regular behaviour.
The structure of the paper is clear. The value of the paper results from appropriate combination of literature studies with the results of an empirical research.
The reviewed paper has a lot of values and has a scientific nature.
In the abstract of the article, information about the purpose and about the applied method should be included, and the period in which the study was conducted ought to be indicated.
Thank reviewer for your suggestions and reminders. We modified the text locally to : additional remarks from line 15 to line 20. Additional contents:(The study used SPSS software version 22.0 and AMOS software version 22.0 to evaluate the data collected. By convenience sampling, it distributed questionnaires to 470 subjects, and collected 470 copies, with a return rate of 100%. After eliminating the invalid samples, there were 453 valid samples, with a valid return rate of 96.3%. We distributed questionnaires at outside the fitness clubs in Taichung City from May 20th to June 13th, 2016.) (In blue)
